

# Effects of high-impact jumping versus resistance exercise on bone mineral content in children and adolescents: a systematic review and meta-analysis

Tingting Miao[1,*], Xun Li[2,*], Wenhua Zhang[1], Fengying Yang[2] and Xiaoqiang Wang[2]

[1] School of Graduate Education, Shandong Sport University, Jinan, Shandong, China
[2] School of Sport and Health, Shandong Sport University, Jinan, Shandong, China
[*] These authors contributed equally to this work.

Corresponding author
Xiaoqiang Wang, wangxiao-qiang_001@126.com

## ABSTRACT

**Background**. The childhood and adolescent period represent a critical window for bone development. Mechanical loading through physical exercise effectively stimulates osteogenic responses, promoting peak bone mass accumulation—a key factor in osteoporosis prevention.

**Objective**. To compare the effects of high-impact jumping versus resistance exercise on bone mineral content (BMC) in children and adolescents, thereby identifying the most osteogenic exercise modality.

**Methods**. We systematically searched PubMed, The Cochrane Library, Web of Science, and Embase from inception to April 4, 2025 for randomized and non-randomized controlled trials investigating high-impact or resistance exercise effects on pediatric BMC. Study quality was assessed using Cochrane's risk-of-bias tool. Meta-analyses were conducted using RevMan 5.4 and Stata 17. To assess robustness, we performed sex-stratified subgroup analyses and sensitivity analyses. Meta-regression with robust variance estimation (RVE) was conducted using the robumeta package.

**Results**. A total of twelve studies involving 940 participants were included. The meta-analysis indicated that high-impact jumping significantly improved BMC in children and adolescents at the lumbar spine (MD = 0.86, 95% CI [0.27–1.45], $p = 0.004$) and femoral neck (MD = 0.11, 95% CI [0.04–0.18], $p = 0.001$). Subgroup analyses by sex demonstrated particularly pronounced improvements in girls, with significant increases in BMC at both the lumbar spine (MD = 1.40, 95% CI [0.16–2.63], $p = 0.03$) and femoral neck (MD = 0.11, 95% CI [0.00–0.21], $p = 0.04$).

**Conclusion**. This study demonstrates that high-impact jumping significantly improves lumbar spine and femoral neck BMC in children and adolescents, with particularly pronounced effects observed in girls. In contrast, resistance exercise did not yield statistically significant improvements in BMC, possibly due to the limited number of studies and methodological limitations. Future research should focus on high-quality randomized controlled trials to inform and optimize bone health interventions for children and adolescents.

## INTRODUCTION

Osteoporosis (OP) is a metabolic bone disorder characterized by decreased bone mineral density (BMD), deterioration of bone microarchitecture and increased bone fragility (*Adejuyigbe et al., 2023*). This condition significantly elevates fracture risk by over 40% in elderly populations and postmenopausal women (*Aibar-Almazán et al., 2022*; *Wang et al., 2020*). Globally, osteoporosis affects over 200 million individuals, with China accounting for approximately 83.9 million cases, making it a critical public health challenge in aging societies (*Tobeiha et al., 2020*; *Zhu & Zheng, 2021*). Research evidence shows that the maximum bone mass achieved during skeletal growth and development, known as peak bone mass (PBM), is an important predictor of osteoporosis risk and fracture incidence (*Hernandez, Beaupré & Carter, 2003*). An increase of 10% in PBM can delay the onset age of OP by 13 years and reduce fracture risk by 50% in the majority of the population (*Kralick & Zemel, 2020*). Although PBM is typically achieved around the age of 30, women accumulate approximately 90% of their PBM by age 18, while men reach the same level by age 20 (*Ishikawa et al., 2013*). Thus, childhood and adolescence constitute a critical window for bone mass accrual, and optimizing skeletal development during this phase is vital for preventing future bone loss and osteoporosis-related complications.

During childhood and adolescence, skeletal growth is primarily driven by bone modeling, characterized by rapid periosteal expansion to increase bone area followed by subsequent mineral deposition, which enhances bone mineral content (BMC). During this period, bones exhibit high sensitivity to mechanical loading, and appropriate exercise interventions during puberty can significantly stimulate bone formation. According to Wolff's Law, bones respond to mechanical loading through mechanotransduction: high-impact forces induce bone tissue deformation, activating osteocyte mechanoreceptors (such as integrins and primary cilia), thereby regulating osteoblast activity and promoting bone formation (*Buck & Stains, 2024*; *Chang, Xu & Zhang, 2022*; *Hart et al., 2020*; *Okubo et al., 2017*; *Uda et al., 2017*). Most meta-analyses have investigated the effects of weight-bearing exercise on bone health in children and adolescents (*Behringer et al., 2014*; *Ishikawa et al., 2013*; *Nogueira, Weeks & Beck, 2014a*; *Specker, Thiex & Sudhagoni, 2015*; *Tan et al., 2014*). Weight-bearing activities refer to exercises that impose mechanical stimuli exceeding daily activity levels on bones, such as jumping and resistance exercise. Meanwhile, the American College of Sports Medicine (ACSM) emphasizes that high-intensity weight-bearing exercises, including impact sports and resistance training, are particularly beneficial for bone mineral accrual in children and adolescents (*Kohrt et al., 2004*).

However, no study has yet directly compared high-impact jumping and resistance exercise to determine which type of exercise is more beneficial for bone mass development in children and adolescents, as well as to investigate their specific effects on different skeletal sites. Research has shown that BMC more accurately reflects bone development during

growth and is not significantly affected by bone size or growth stage (*Wren et al., 2005*). Therefore, this systematic review and meta-analysis compares the effects of high-impact jumping *versus* resistance exercise on lumbar spine, femoral neck, and whole-body BMC in adolescents, aiming to provide evidence-based recommendations for optimizing exercise interventions to enhance skeletal development.

## MATERIALS & METHODS

### Protocol and registration

This systematic review and meta-analysis were conducted in accordance with the Preferred Reporting Items for Systematic Reviews and Meta-Analyses statement guidelines (*Page et al., 2021*). The study protocol was registered in the International Prospective Register of Systematic Reviews (ID: CRD 42024625921).

### Search strategy

A literature search was conducted by two researchers, Zhang and Yang, based on the inclusion and exclusion criteria, which included PubMed, Embase, The Cochrane Library and Web of Science databases, spanned from the establishment of each database to April 4, 2025, without any language restriction. In case of disagreement, the researchers will resolve the issue through discussion with the first author, and if the disagreement is still not agreed upon, the first author will serve as the final adjudicator. The PubMed database was systematically searched employing the following criteria: (((((((((("Exercise"[Mesh]) OR ("Resistance Training"[Mesh])) OR (sports)) OR (High-impact sports)) OR (Jump)) OR (physical activity)) OR (training))AND (((((((("Adolescent"[Mesh]) OR (student)) OR (Puberty)) OR (children)) OR (kids)) OR (child)) OR (pediatrics))) AND ((((bone) OR (bone health)) OR (Bone mineral density)) OR (Bone mineral content)) AND (clinical trial[Filter] OR controlled clinical trial [Filter]OR randomized controlled trial [Filter]).

### Inclusion and exclusion criteria for the studies

The inclusion criteria followed the PICOS framework: (a) participants: healthy children and adolescents (<18 years) without hepatic, renal, endocrine, or other metabolic bone diseases; (b) intervention: structured high-impact exercise or resistance training programs; (c) control: routine school physical education without supplemental exercise interventions; (d) outcomes: $\Delta$BMC (pre-to-post intervention) measured by dual-energy X-ray absorptiometry (DXA) at lumbar spine, femoral neck and whole-body (studies using dual-photon absorptiometry were excluded, *e.g.*, *Blimkie et al. (1996)*); (e) study design: randomized or non-randomized controlled trials.

Exclusion criteria comprised: (a) participants $\geq$ 18 years; (b) concurrent pharmacotherapy; (c) non-extractable outcome data; (d) abstract-only/review articles; (e) animal studies.

### Literature screening and data extraction

Two researchers independently screened the literature and extracted data according to the predefined inclusion/exclusion criteria, with any discrepancies resolved through consensus
discussion involving a third reviewer. The systematic search was conducted across multiple databases using standardized search strategies. All retrieved records were imported into EndNote for duplicate removal, followed by title/abstract screening to exclude irrelevant studies. The extracted study characteristics included: (a) Basic characteristics: first author, publication year, country, study design, sample size, participants' age and gender distribution; (b) Intervention characteristics: exercise modality (specific description), intervention duration, intervention frequency, intervention period and outcome measures.

## Quality assessment

Two researchers independently evaluated the methodological quality of included studies using the Cochrane Collaboration's Risk of Bias tool (*Higgins et al., 2011*), assessing seven domains: (a) random sequence generation (selection bias); (b) allocation concealment (selection bias); (c) blinding of participants and personnel (performance bias); (d) blinding of outcome assessment (detection bias); (e) incomplete outcome data (attrition bias); (f) selective reporting (reporting bias); (g) other potential biases. Each domain was judged as 'low risk', 'high risk' or 'unclear risk' of bias.

## Data analysis

Meta-analysis was conducted using RevMan 5.4 and Stata 17 software. As all included studies measured BMC in grams (g) by Dual-energy X-ray Absorptiometry (DXA), the mean difference (MD) with 95% confidence intervals (95% CI) was selected as the effect measure. We preferentially extracted within-group change scores (follow-up minus baseline values) and their standard deviations (SDs) for both intervention and control groups. When studies did not directly report SDs of change scores, we calculated them using the following formula:

$$\text{SD}_{change} = \sqrt{\text{SD}^2_{baseline} + \text{SD}^2_{follow-up} - 2 \times r \times \text{SD}_{baseline} \times \text{SD}_{follow-up}}$$

a default correlation coefficient ($r$) of 0.5 was used when unreported. Heterogeneity was evaluated using Cochrane's Q-test ($P < 0.10$) and $I^2$ statistics, with interpretation thresholds set as: 0%–40% (might not be significant), 30%–60% (moderate), 50%–90% (substantial), and 75%–100% (considerable) according to Cochrane Handbook guidelines (*Cochrane Collaboration, 2024*). A random-effects model was applied when $I^2 \geq 50\%$. To ensure result robustness, we performed subgroup analyses (by sex), sensitivity analyses. Meta-regression with robust variance estimation (RVE) was performed using the robumeta package, as described by *Fisher & Tipton (2015)*, to evaluate the effects of moderator variables (duration, frequency, and period) on BMC at the lumbar spine, femoral neck and whole-body in jump and resistance exercises.

# RESULTS

## Search results

A total of 7,368 relevant studies were initially identified, including 1,724 from the PubMed database, 3,115 from Embase, 1,508 from The Cochrane Library and 1,021 from Web of Science. These studies were exported to EndNote and after removing duplicates, 3,403
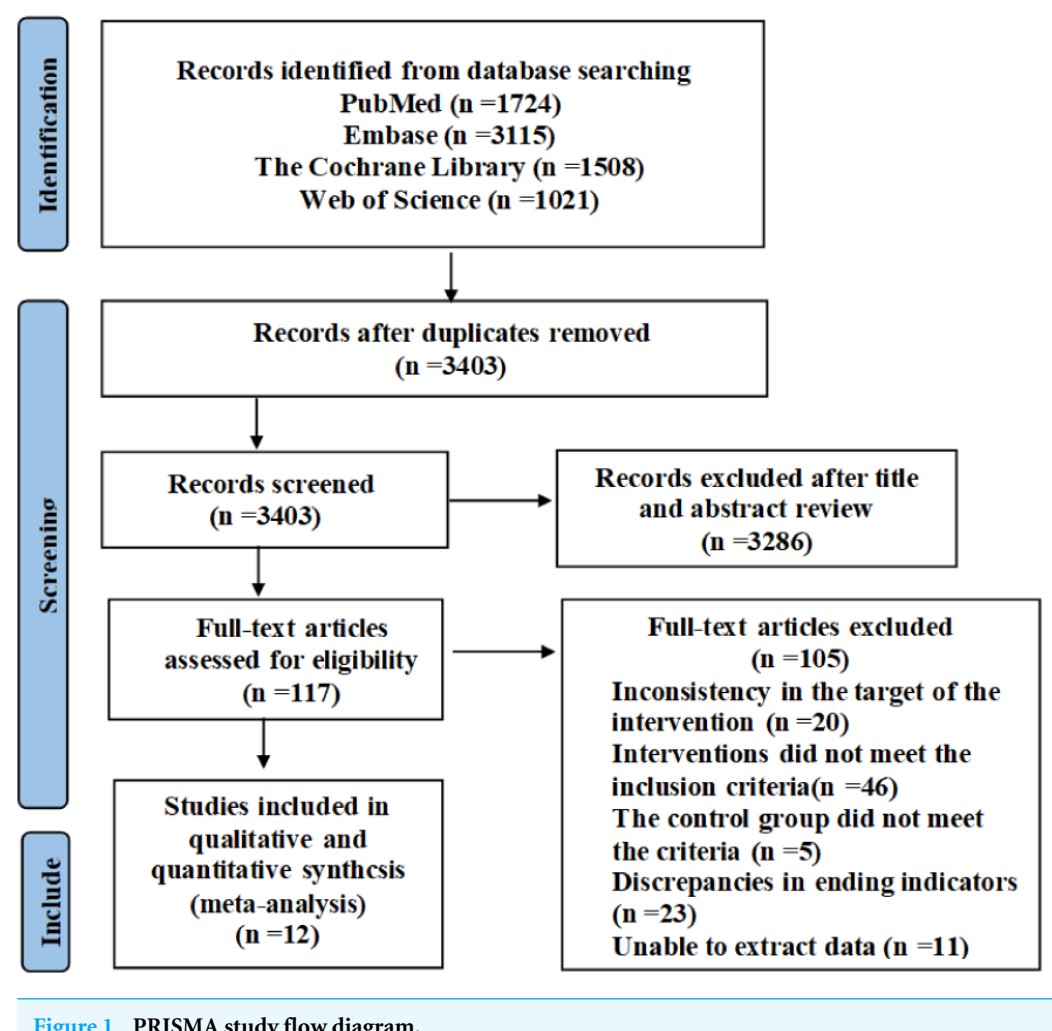

**Figure 1  PRISMA study flow diagram.**

articles remained. After screening titles and abstracts, 3,286 articles were excluded, leaving 117 studies for full-text review to assess eligibility for inclusion. Among them, inconsistency in the target of the intervention ($n = 20$); interventions did not meet the inclusion criteria ($n = 46$); the control group did not meet the criteria ($n = 5$); discrepancies in ending indicators ($n = 23$); unable to extract data ($n = 11$), and finally the remaining 12 articles were included in meta-analysis. Figure 1 shows the specific process.

## Basic characteristics of the included studies

The meta-analysis included 12 studies (*Arnett & Lutz, 2002*; *Dowthwaite et al., 2019*; *Fuchs, Bauer & Snow, 2001*; *Gómez et al., 2021*; *Macdonald et al., 2008*; *Nichols et al., 2008*; *Nichols, Sanborn & Love, 2001*; *Nogueira, Weeks & Beck, 2014b*; *Nogueira, Weeks & Beck, 2015*; *Thein-Nissenbaum et al., 2023*; *Weeks, Young & Beck, 2008*; *Witzke & Snow, 2000*), published between 2000–2023, involving 940 total participants (intervention group: $n = 524$; control group: $n = 416$), aged 7.3 (*Fuchs, Bauer & Snow, 2001*)–16.8 (*Gómez et al., 2021*) years (mean age: females 12.6; males 11.1). Sample sizes in intervention

**Table 1 Basic characteristics of the included studies.**

| Author, Year | Country | Design | Sample size | | Age | | Sex |
|---|---|---|---|---|---|---|---|
| | | | EX | C | EX | C | |
| Arnett, 2002 (HI) | USA | RCT | 13 | 12 | 14.9 ± 0.6 | 14.8 ± 0.9 | Female |
| Anett, 2002 (LO) | USA | RCT | 12 | 12 | 14.6 ± 0.7 | 14.8 ± 0.9 | Female |
| Dowthwaite, 2019 (HI) | USA | Non-RCT | 19 | 21 | 13.2 ± 0.2 | 13.2 ± 0.3 | Female |
| Dowthwaite, 2019 (LO) | USA | Non-RCT | 22 | 21 | 13.1 ± 0.3 | 13.2 ± 0.3 | Female |
| Fuchs, 2001 | USA | RCT | 45 | 44 | 7.5 ± 0.16 | 7.6 ± 0.17 | Male/ Female |
| Gómez, 2020 | USA | Non-RCT | 15 | 16 | 15.4 ± 1.4 | 15.4 ± 1.2 | Female |
| Macdonald, 2008 (boy) | Canada | RCT | 66 | 58 | 10.2 ± 0.5 | 10.3 ± 0.7 | Male |
| Macdonald, 2008 (girl) | Canada | RCT | 43 | 55 | 10.2 ± 0.6 | 10.2 ± 0.5 | Female |
| Nichols, 2001 | USA | RCT | 5 | 11 | 16.01 ± 0.3 | 15.5 ± 0.2 | Female |
| Nichols, 2008 | USA | RCT | 61 | 28 | 9.7 ± 0.3 | 9.7 ± 0.5 | Male/ Female |
| Thein-Nissenbaum, 2023 (HI) | USA | Non-RCT | 25 | 23 | 11.6 ± 0.3 | 11.7 ± 0.3 | Female |
| Thein-Nissenbaum, 2023 (LO) | USA | Non-RCT | 20 | 23 | 11.5 ± 0.3 | 11.5 ± 0.3 | Female |
| Nogueira, 2014 | Australia | RCT | 71 | 67 | 10.5 ± 0.6 | 10.7 ± 0.6 | Female |
| Nogueira, 2015 | Australia | RCT | 30 | 6 | 10.5 ± 0.5 | 10.7 ± 0.6 | Male |
| Weeks, 2008 (boy) | Australia | RCT | 22 | 24 | 13.8 ± 0.4 | 13.8 ± 0.4 | Male |
| Weeks, 2008 (girl) | Australia | RCT | 30 | 23 | 13.7 ± 0.4 | 13.7 ± 0.5 | Female |
| Witzke, 2000 | USA | Non-RCT | 25 | 28 | 14.6 ± 0.4 | 14.5 ± 0.6 | Female |

**Notes.**
Arnett & Lutz, 2002; Dowthwaite et al., 2019; Fuchs, Bauer & Snow, 2001; Gómez et al., 2021; Macdonald et al., 2008; Nichols, Sanborn & Love, 2001; Nichols et al., 2008; Thein-Nissenbaum et al., 2023; Nogueira, Weeks & Beck, 2015; Nogueira, Weeks & Beck, 2014a; Nogueira, Weeks & Beck, 2014b; Weeks, Young & Beck, 2008; Witzke & Snow, 2000.

groups ranged from 5 (*Nichols, Sanborn & Love, 2001*) to 71 (*Nogueira, Weeks & Beck, 2014b*) participants. Studies were conducted in the United States ($n = 8$) (*Arnett & Lutz, 2002*; *Dowthwaite et al., 2019*; *Fuchs, Bauer & Snow, 2001*; *Gómez et al., 2021*; *Nichols et al., 2008*; *Nichols, Sanborn & Love, 2001*; *Thein-Nissenbaum et al., 2023*; *Witzke & Snow, 2000*), Australia ($n = 3$) (*Nogueira, Weeks & Beck, 2014b*; *Nogueira, Weeks & Beck, 2015*; *Weeks, Young & Beck, 2008*) and Canada ($n = 1$) (*Macdonald et al., 2008*). Detailed baseline characteristics are presented in Table 1.

## Characteristics of exercise interventions included in the studies

Eight studies (*Arnett & Lutz, 2002*; *Fuchs, Bauer & Snow, 2001*; *Macdonald et al., 2008*; *Nichols et al., 2008*; *Nogueira, Weeks & Beck, 2014b*; *Nogueira, Weeks & Beck, 2015*; *Weeks, Young & Beck, 2008*; *Witzke & Snow, 2000*) implemented high-impact exercise, while four studies (*Dowthwaite et al., 2019*; *Gómez et al., 2021*; *Nichols, Sanborn & Love, 2001*; *Thein-Nissenbaum et al., 2023*) involved resistance exercise. The intervention duration ranged from five (*Arnett & Lutz, 2002*) to 60 minutes (*Gómez et al., 2021*), with a frequency of two (*Nichols et al., 2008*; *Weeks, Young & Beck, 2008*) to five sessions per week (*Macdonald et al., 2008*). The intervention period spanned three (*Gómez et al., 2021*) to 24 months (*Dowthwaite et al., 2019*). BMC outcomes were measured at three sites: lumbar spine

**Table 2** Characteristics of exercise interventions included in the studies.

| Author, Year | Exercise modality | Description | Duration (min) | Frequency (week) | Period (month) | BMC (g) |
|---|---|---|---|---|---|---|
| Arnett, 2002 (HI) | High-impact | Rope jumping + weighted vest | 10 | 4 | 4 | LS, FN |
| Arnett, 2002 (LO) | High-impact | Rope jumping + weighted vest | 5 | 4 | 4 | LS, FN |
| Dowthwaite, 2019 (HI) | Resistance | Elastic bands, handheld weights, medicine balls, bodyweight | 8–12 | 2–3 | 24 | LS, FN, WB |
| Dowthwaite, 2019 (LO) | Resistance | Elastic bands, handheld weights, medicine balls, bodyweight | 8–12 | 2–3 | 24 | LS, FN, WB |
| Fuchs, 2001 | High-impact | Jumping | 20 | 3 | 7 | LS, FN |
| Gómez, 2020 | Resistance | Free weights (dumbbells, barbells) + Cybex stack | 60 | 3 | 3 | WB |
| Macdonald, 2008 (boy) | High-impact | Jumping | 15 | 5 | 16 | LS, FN, WB |
| Macdonald, 2008 (girl) | High-impact | Jumping | 15 | 5 | 16 | LS, FN, WB |
| Nichols, 2001 | Resistance | Free weights + machines | 30–45 | 3 | 15 | LS, FN, WB |
| Nichols, 2008 | High-impact | Jumping | 6–8 | 2 | 20 | LS, FN, WB |
| Nissenbaum, 2023 (HI) | Resistance | Elastic bands, handheld weights, multi-planar bodyweight movements | 8–12 | 2–3 | 6 | LS, FN, WB |
| Nissenbaum, 2023 (LO) | Resistance | Elastic bands, handheld weights, multi-planar bodyweight movements | 8–12 | 2–3 | 6 | LS, FN, WB |
| Nogueira, 2014 | High-impact | Jumping+ Capoeira | 10 | 3 | 9 | LS, FN, WB |
| Nogueira, 2015 | High-impact | Jumping+ Capoeira | 10 | 3 | 9 | LS, FN, WB |
| Weeks, 2008 (boy) | High-impact | Jumping + rope skipping | 10 | 2 | 8 | LS, FN, WB |
| Weeks, 2008 (girl) | High-impact | Jumping + rope skipping | 10 | 2 | 8 | LS, FN, WB |
| Witzke, 2000 | High-impact | Resistance (first 3 mo) + jumping (last 6 mo) | 30–45 | 3 | 9 | LS, FN, WB |

**Notes.**
HI, High intensity; LO, Low intensity; LS, lumbar spine; FN, femoral neck; WB, whole-body.

*Arnett & Lutz, 2002*; *Dowthwaite et al., 2019*; *Fuchs, Bauer & Snow, 2001*; *Gómez et al., 2021*; *Macdonald et al., 2008*; *Nichols, Sanborn & Love, 2001*; *Nichols et al., 2008*; *Thein-Nissenbaum et al., 2023*; *Nogueira, Weeks & Beck, 2015*; *Nogueira, Weeks & Beck, 2014a*; *Nogueira, Weeks & Beck, 2014b*; *Weeks, Young & Beck, 2008*; *Witzke & Snow, 2000*.

($n = 11$), femoral neck ($n = 11$) and whole-body ($n = 10$). Detailed intervention characteristics are presented in Table 2.

### Risk of bias assessment of included studies

According to the Cochrane risk of bias assessment criteria, the methodological quality of included studies demonstrated the following characteristics: Four studies (*Dowthwaite et al., 2019*; *Gómez et al., 2021*; *Thein-Nissenbaum et al., 2023*; *Witzke & Snow, 2000*) used non-randomized grouping (by school/class), while one study (*Nichols, Sanborn & Love, 2001*) showed significant attrition bias (85% dropout in intervention group) and low adherence (mean compliance: 73%). Two studies (*Arnett & Lutz, 2002*; *Gómez et al., 2021*) had short intervention periods (3–4 months), potentially limiting long-term effect observation. Although no studies reported allocation concealment or blinding

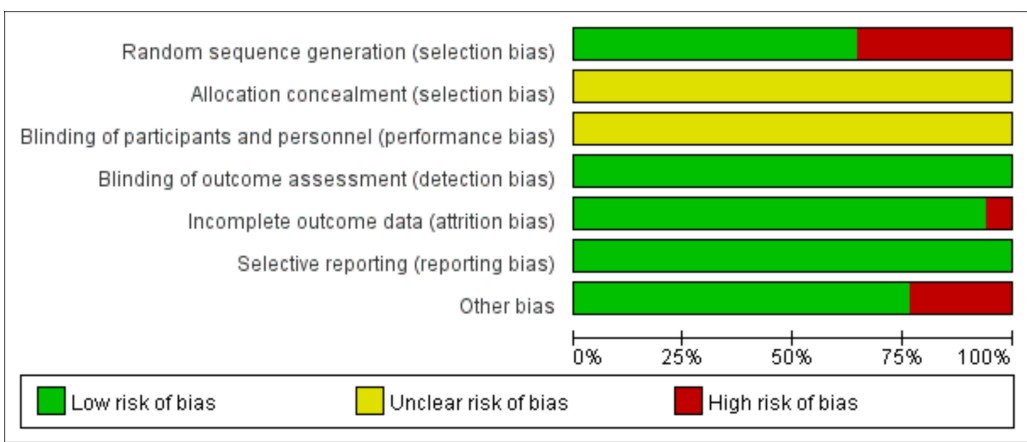

**Figure 2   Risk of bias of the included studies.**

of participants/personnel, all utilized DXA for objective outcome measurement, resulting in low detection bias and no evidence of selective reporting bias. The risk of bias evaluation results are shown in Figs. 2 and 3.

## Meta-analysis results
### *Effects of high-impact jumping on lumbar spine BMC*
Eight studies (*Arnett & Lutz, 2002*; *Fuchs, Bauer & Snow, 2001*; *Macdonald et al., 2008*; *Nichols et al., 2008*; *Nogueira, Weeks & Beck, 2014b*; *Nogueira, Weeks & Beck, 2015*; *Weeks, Young & Beck, 2008*; *Witzke & Snow, 2000*) evaluated the effects of jumping on lumbar spine BMC. No significant heterogeneity was detected ($P = 0.76$, $I^2 = 0\%$), and a fixed-effects model was applied. The meta-analysis demonstrated a statistically significant increase in lumbar spine BMC changes in the exercise group compared to controls (MD = 0.86, 95% CI [0.27–1.45], $P = 0.004$) (Fig. 4).

### Effects of high-impact jumping on femoral neck BMC
Eight studies (*Arnett & Lutz, 2002*; *Fuchs, Bauer & Snow, 2001*; *Macdonald et al., 2008*; *Nichols et al., 2008*; *Nogueira, Weeks & Beck, 2014b*; *Nogueira, Weeks & Beck, 2015*; *Weeks, Young & Beck, 2008*; *Witzke & Snow, 2000*) examined the effects of jumping on femoral neck BMC. No significant heterogeneity was observed ($P = 0.93$, $I^2 = 0\%$), warranting a fixed-effects model. Meta-analysis revealed a statistically significant increase in femoral neck BMC changes in the exercise group compared to controls (MD = 0.11, 95% CI [0.04–0.18], $P = 0.001$) (Fig. 5).

### Effects of high-impact jumping on whole-body BMC
Six studies (*Macdonald et al., 2008*; *Nichols et al., 2008*; *Nogueira, Weeks & Beck, 2014b*; *Nogueira, Weeks & Beck, 2015*; *Weeks, Young & Beck, 2008*; *Witzke & Snow, 2000*) evaluated the effects of jumping on whole-body BMC. No significant heterogeneity was observed ($P = 0.89$, $I^2 = 0\%$) and a fixed-effects model was employed. The meta-analysis found no statistically significant difference in whole-body BMC changes between exercise and control groups (MD = 5.11, 95% CI [−42.18–52.40], $P = 0.83$) (Fig. 6).

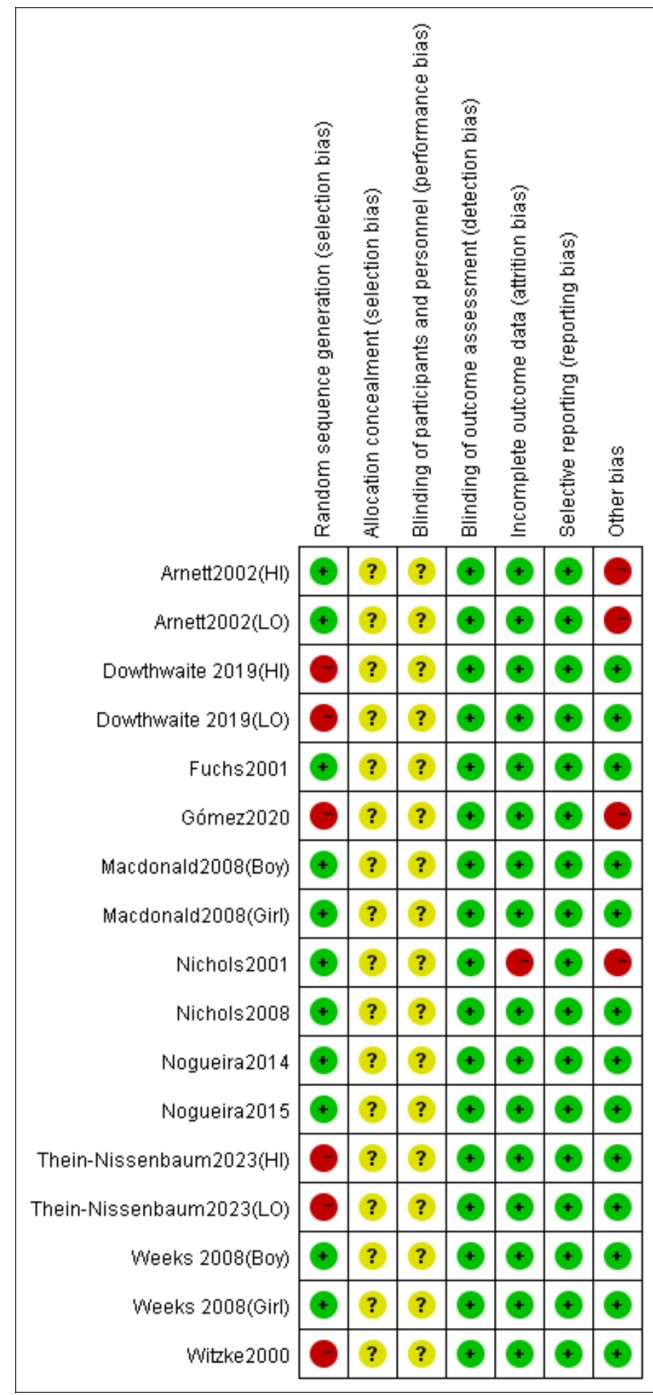

**Figure 3** **Risk of bias summary of the included studies.** Note. *Arnett & Lutz, 2002*; *Dowthwaite et al., 2019*; *Fuchs, Bauer & Snow, 2001*; *Gómez et al., 2021*; *Macdonald et al., 2008*; *Nichols, Sanborn & Love, 2001*; *Nichols et al., 2008*; *Thein-Nissenbaum et al., 2023*; *Nogueira, Weeks & Beck, 2015*; *Nogueira, Weeks & Beck, 2014a*; *Nogueira, Weeks & Beck, 2014b*; *Weeks, Young & Beck, 2008*; *Witzke & Snow, 2000*.

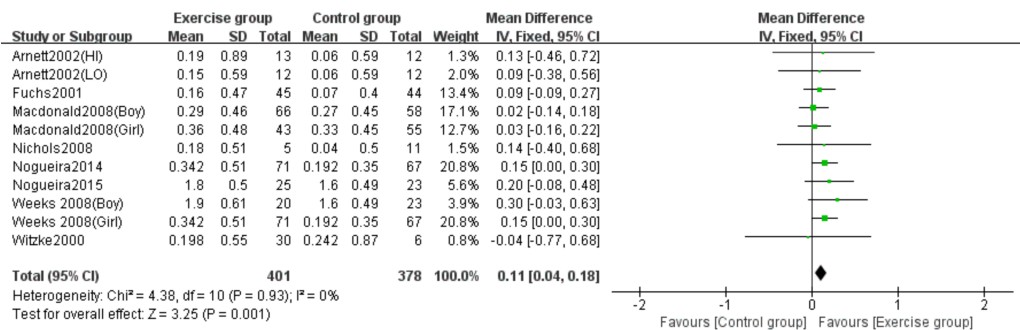

**Figure 4** **Forest plot of the meta-analysis on the effects of high-impact jumping on lumbar spine BMC.** Note. *Arnett & Lutz, 2002*; *Fuchs, Bauer & Snow, 2001*; *Macdonald et al., 2008*; *Nichols et al., 2008*; *Nogueira, Weeks & Beck, 2015*; *Nogueira, Weeks & Beck, 2014a*; *Nogueira, Weeks & Beck, 2014b*; *Weeks, Young & Beck, 2008*; *Witzke & Snow, 2000*.

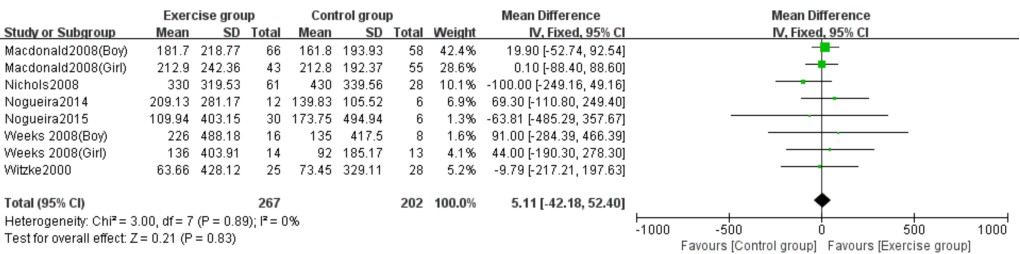

**Figure 5** **Forest plot of the meta-analysis on the effects of high-impact jumping on femoral neck BMC.** Note. *Arnett & Lutz, 2002*; *Fuchs, Bauer & Snow, 2001*; *Macdonald et al., 2008*; *Nichols et al., 2008*; *Nogueira, Weeks & Beck, 2015*; *Nogueira, Weeks & Beck, 2014a*; *Nogueira, Weeks & Beck, 2014b*; *Weeks, Young & Beck, 2008*; *Witzke & Snow, 2000*.

**Figure 6** **Forest plot of the meta-analysis on the effects of high-impact jumping on whole- body BMC.** Note. *Macdonald et al., 2008*; *Nichols et al., 2008*; *Nogueira, Weeks & Beck, 2015*; *Nogueira, Weeks & Beck, 2014a*; *Nogueira, Weeks & Beck, 2014b*; *Weeks, Young & Beck, 2008*; *Witzke & Snow, 2000*.

## Effects of resistance exercise on lumbar spine BMC

Three studies (*Dowthwaite et al., 2019*; *Nichols, Sanborn & Love, 2001*; *Thein-Nissenbaum et al., 2023*) examined the effect of resistance exercise on lumbar spine BMC. No significant

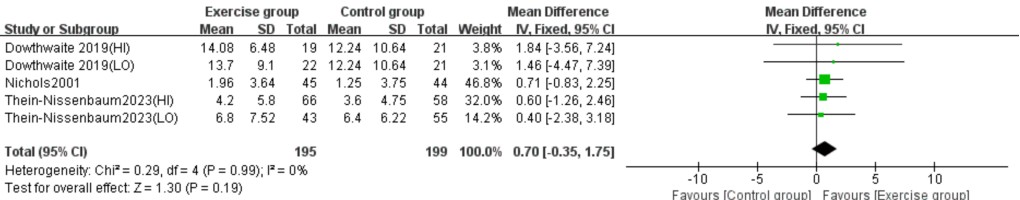

**Figure 7** Forest plot of the meta-analysis on the effects of resistance exercise on lumbar spine BMC.
Note. *Dowthwaite et al., 2019*; *Nichols, Sanborn & Love, 2001*; *Thein-Nissenbaum et al., 2023*.

**Figure 8** Forest plot of the meta-analysis on the effects of resistance exercise on femoral neck BMC.
Note. *Dowthwaite et al., 2019*; *Nichols, Sanborn & Love, 2001*; *Thein-Nissenbaum et al., 2023*.

heterogeneity was detected ($P = 0.99$, $I^2 = 0\%$) and a fixed-effects model was applied. The meta-analysis showed no statistically significant difference in lumbar spine BMC changes between exercise and control groups (MD = 0.70, 95% CI [$-0.35$–$1.75$], $P = 0.19$) (Fig. 7).

## Effects of resistance exercise on femoral neck BMC

Three studies (*Dowthwaite et al., 2019*; *Nichols, Sanborn & Love, 2001*; *Thein-Nissenbaum et al., 2023*) evaluated the effect of resistance exercise on femoral neck BMC. No significant heterogeneity was observed ($P = 0.92$, $I^2 = 0\%$) and a fixed-effects model was used. The meta-analysis revealed no statistically significant difference in femoral neck BMC changes between exercise and control groups (MD = 0.06, 95% CI [$-0.04$–$0.15$], $P = 0.25$) (Fig. 8).

## Effects of resistance exercise on whole-body BMC

Four studies (*Dowthwaite et al., 2019*; *Gómez et al., 2021*; *Nichols, Sanborn & Love, 2001*; *Thein-Nissenbaum et al., 2023*) investigated the effect of resistance exercise on whole-body BMC. The analysis showed no significant heterogeneity ($P = 0.92$, $I^2 = 0\%$), and a fixed-effects model was employed. Meta-analysis results indicated no statistically significant difference in whole-body BMC changes between exercise and control groups (MD = 56.22, 95% CI [$-3.98$–$116.43$], $P = 0.07$) (Fig. 9).

## Subgroup analysis results

The subgroup analysis by sex revealed differential effects of high-impact jumping on BMC. Three studies involving boys (*Macdonald et al., 2008*; *Nogueira, Weeks & Beck, 2015*; *Weeks, Young & Beck, 2008*) and five studies involving girls (*Arnett & Lutz, 2002*; *Macdonald et al.,*

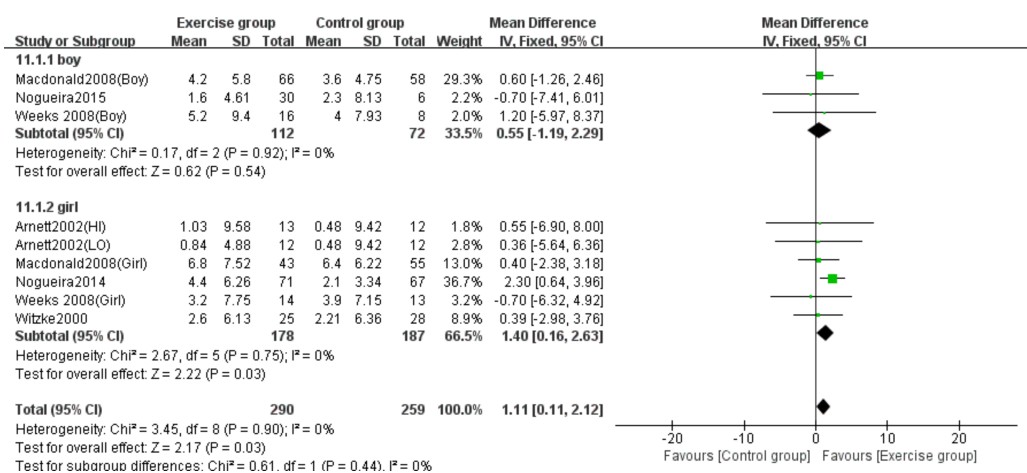

**Figure 9** Forest plot of the meta-analysis on the effects of resistance exercise on whole-body BMC. Note. *Dowthwaite et al., 2019*; *Gómez et al., 2021*; *Nichols, Sanborn & Love, 2001*; *Thein-Nissenbaum et al., 2023*.

**Figure 10** Forest plot of the subgroup analysis on the effects of high-impact jumping on lumbar spine BMC. Note. *Arnett & Lutz, 2002*; *Macdonald et al., 2008*; *Nogueira, Weeks & Beck, 2015*; *Nogueira, Weeks & Beck, 2014a*; *Nogueira, Weeks & Beck, 2014b*; *Weeks, Young & Beck, 2008*; *Witzke & Snow, 2000*.

*2008*; *Nogueira, Weeks & Beck, 2014b*; *Weeks, Young & Beck, 2008*; *Witzke & Snow, 2000*) were analyzed for lumbar spine BMC. Subgroup analysis demonstrated that jumping significantly increased lumbar spine BMC in girls compared to controls (MD = 1.40, 95% CI [0.16–2.63], $P = 0.03$), while no significant effect was observed in boys (MD = 0.55, 95% CI [−1.19–2.29], $P = 0.54$) (Fig. 10).

Three studies (*Macdonald et al., 2008*; *Nogueira, Weeks & Beck, 2015*; *Weeks, Young & Beck, 2008*) examined the effects on femoral neck BMC in boys, while five studies (*Arnett & Lutz, 2002*; *Macdonald et al., 2008*; *Nogueira, Weeks & Beck, 2014b*; *Weeks, Young & Beck, 2008*; *Witzke & Snow, 2000*) evaluated girls. Subgroup analysis demonstrated that jumping significantly increased femoral neck BMC in girls compared to controls (MD = 0.11, 95% CI [0.00–0.21], $P = 0.04$), but showed no significant effect in boys (MD = 0.02, 95% CI [−0.13–0.17], $P = 0.79$) (Fig. 11).

Three studies (*Macdonald et al., 2008*; *Nogueira, Weeks & Beck, 2015*; *Weeks, Young & Beck, 2008*) evaluated the effects on whole-body BMC in boys, while four studies (*Macdonald et al., 2008*; *Nogueira, Weeks & Beck, 2014b*; *Weeks, Young & Beck, 2008*;

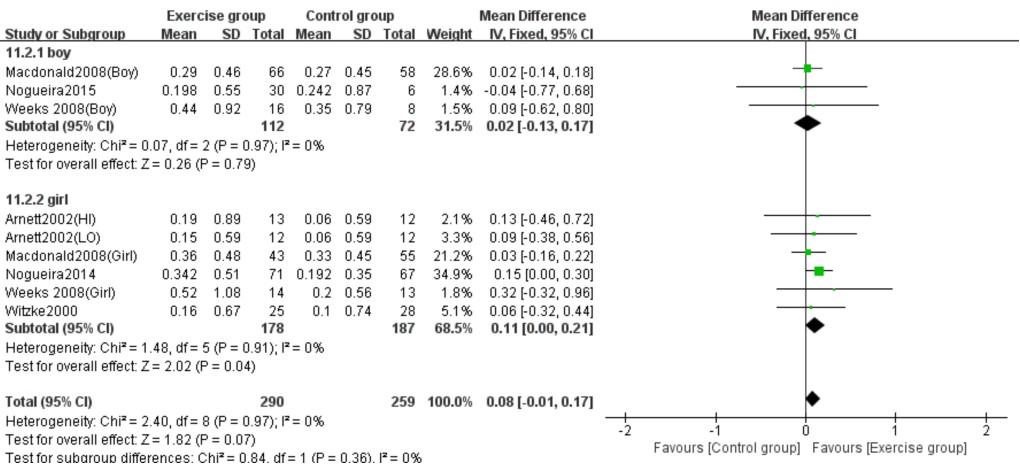

**Figure 11 Forest plot of the subgroup analysis on the effects of high-impact jumping on femoral neck BMC.** Note. *Arnett & Lutz, 2002*; *Macdonald et al., 2008*; *Nogueira, Weeks & Beck, 2015*; *Nogueira, Weeks & Beck, 2014a*; *Nogueira, Weeks & Beck, 2014b*; *Weeks, Young & Beck, 2008*; *Witzke & Snow, 2000*.

**Figure 12 Forest plot of the subgroup analysis on the effects of high-impact jumping on whole-body BMC.** Note. *Macdonald et al., 2008*; *Nogueira, Weeks & Beck, 2015*; *Nogueira, Weeks & Beck, 2014a*; *Nogueira, Weeks & Beck, 2014b*; *Weeks, Young & Beck, 2008*; *Witzke & Snow, 2000*.

*Witzke & Snow, 2000*) assessed girls. The subgroup analysis revealed no statistically significant effects of jumping on whole-body BMC changes in either girls (MD = 13.62, 95% CI [−57.10–84.3], $P$ = 0.71) or boys (MD = 20.06, 95% CI [−50.25–90.38], $P$ = 0.58) compared to control groups (Fig. 12).

All resistance exercise studies exclusively involved female participants. As previously reported in the meta-analysis, resistance exercise demonstrated no statistically significant effects on lumbar spine, femoral neck or whole-body BMC in this population.

## Sensitivity analysis

The sensitivity analysis confirmed the stability of effect sizes for both high-impact jumping and resistance exercise on femoral neck, lumbar spine and whole-body BMC. The pooled

effect sizes (95% CI) remained robust without significant alterations upon sequential exclusion of individual studies Fig. S1.

### Meta-regression analysis

The robust meta-regression analysis revealed distinct patterns of association between exercise parameters and BMC across anatomical sites. For jumping, longer period showed a significant negative association with whole-body BMC ($\beta = -12.74$, $P = 0.028$), while no significant effects were observed at lumbar spine or femoral neck sites (all $P > 0.05$). In resistance exercise, both longer duration ($\beta = -1.68$, $P = 0.032$) and period length ($\beta = -3.83$, $P = 0.014$) were negatively associated with whole-body BMC, with consistent null findings at other anatomical sites Table S1.

## DISCUSSION

This systematic review and meta-analysis included 12 randomized and non-randomized controlled trials comparing the effects of high-impact jumping and resistance exercise on BMC in children and adolescents, focusing on the lumbar spine, femoral neck and whole-body BMC. The results indicated that high-impact jumping significantly outperformed resistance exercise in improving BMC in the lumbar spine and femoral neck, while no significant difference was found in the effect on whole-body BMC. Subgroup analysis further revealed that the beneficial effects of high-impact jumping on lumbar spine and femoral neck BMC were more pronounced in girls compared to boys.

High-impact exercise has been shown to exert significant site-specific effects on bone stimulation. As noted in the study by *Kato et al. (2006)*, high-impact exercise has a positive effect on weight-bearing axial and appendicular bones. When athletes begin training in early adolescence, the adaptive response of the bones to exercise load is more pronounced. *Haapasalo et al. (1998)* found that female tennis players exhibited a significantly greater increase in BMD in their dominant arm compared to their non-dominant arm (proximal humerus, humeral shaft, and distal radius), further confirming the site-specific effects of exercise load. Additionally, the study by *Vlachopoulos et al. (2017)* found that adolescent athletes participating in high-impact sports, such as soccer, had significantly better bone mass and bone structure at weight-bearing sites, such as the lumbar spine and femoral neck, compared to adolescents engaged in non-impact sports. Consistent with these findings, the results of the present study demonstrate that high-impact jumping significantly enhances BMC in the key weight-bearing sites—lumbar spine and femoral neck—in children and adolescents. Furthermore, the 7-month follow-up data from *Fuchs & Snow (2002)* included in this study indicate that high-impact jumping resulted in a 4% higher BMC in the femoral neck of prepubertal children compared to the control group, while no significant effect was observed in the spine. This difference not only confirms the site-specific nature of exercise stimuli but also suggests that different skeletal sites exhibit distinct long-term responses to exercise interventions, including variations in bone mass changes after cessation of training. Additionally, *Nikander et al. (2010)* proposed that adult athletes engaged in impact sports experience site-specific differences in bone adaptation. For example, the cortical bone at the distal tibia may thicken *via* endosteal remodeling, while the tibial shaft mainly responds

with periosteal apposition. This finding suggests that site-specific changes in bone structure are associated with their underlying biological mechanisms.

Impact exercise refers to rapid and powerful movements generated through the stretch-shortening cycle of muscles, which induce explosive concentric muscle contractions following eccentric muscle actions. The effects of impact exercise on bone vary with intensity. A study by *Maïmoun et al. (2013)* on adolescent female athletes found that high-impact exercise (such as artistic gymnastics) significantly improved BMD at weight-bearing sites like the lumbar spine and femoral neck, while moderate-impact exercise (such as rhythmic gymnastics) only affected the lower limbs, and low-impact exercise (such as swimming) had no significant effect. This study primarily focuses on a common form of high-impact jumping among adolescents, which generates significant ground reaction forces (GRF) that subject the bones to high-intensity and dynamic mechanical loading (*Berro et al., 2024*; *Florence, Oosthuyse & Bosch, 2023*; *Gómez-Bruton et al., 2017*), compared to other moderate or low-impact exercises such as running and brisk walking, jumping is one of the most effective forms of osteogenic stimulation. There are significant differences in the GRF generated by different types of jumping. *McKay et al. (2005)* found that regular jumps generate GRF 2–5 times body weight, while enhanced and reverse jumps generate GRF exceeding 5 times body weight. In this study, the GRF used by *Fuchs, Bauer & Snow (2001)* reached 8.8 times body weight, *Macdonald et al. (2008)* and *Witzke & Snow (2000)* also employed high-GRF reverse and enhanced jumps. These high forces are likely a key factor contributing to the significant effects of jumping.

A 12-month longitudinal study by *Agostinete et al. (2024)* showed that compared to swimmers, resistance exercise increased upper limb and whole-body areal bone mineral density (aBMD) in swimmers. The mechanism underlying this effect is that resistance exercise directly applies mechanical loading through active muscle contractions (isometric, concentric, and eccentric), while also stimulating bone through the secretion of muscle-derived factors such as Irisin and IGF-1, thereby activating osteogenic responses and increasing BMC (*Herrmann et al., 2020*). However, in the present study, resistance exercise had no significant effect on lumbar spine, femoral neck or whole-body BMC, which may be attributable to several factors. First, some studies lacked clear intensity standards, and the resistance exercises for children and adolescents typically used lower intensities due to safety concerns, compounded by this age group's lower exercise tolerance, for instance, *Nichols, Sanborn & Love (2001)* observed high dropout rates and low adherence in their study. Second, the conclusions are supported by only four studies with limited sample sizes, which may have reduced statistical power and thus impacted the reliability of the results. Finally, given the possibility of insufficient exercise intensity in the resistance training protocols, *Min et al. (2019)* recommend combining impact exercise with resistance exercise as a more effective strategy for promoting PBM accumulation in adolescents.

Our findings indicate that high-impact jumping was more effective in enhancing lumbar spine and femoral neck BMC in girls compared to boys. This difference may reflect varied pubertal development timing. The average age of girls in the included studies was 12.6 years, which typically coincides with the period of rapid bone mass accrual, whereas the average age of boys was 11.1 years, a stage when the rapid growth phase is just beginning

(*Baxter-Jones & Jackowski, 2021*). After puberty, males generally attain higher BMC and BMD than females (*Ortona et al., 2023*). *Mackelvie et al. (2001)* highlighted that around the age of 12.5, girls enter a critical window for bone mineral deposition on both periosteal and endosteal surfaces, during which they may be more responsive to mechanical loading. Moreover, the number of studies examining the effects of high-impact jumping on lumbar spine and femoral neck BMC was greater for girls than for boys, which may have contributed to an overestimation of the gender difference and introduced some bias into the findings.

The meta-analysis demonstrated that neither high-impact jumping nor resistance exercise resulted in significant improvements in whole-body BMC. Importantly, robust meta-regression analysis identified a significant negative association between longer intervention periods and whole-body BMC changes in both exercise modalities. This finding may, in part, be attributed to declining adherence over time, as previously reported by *Nichols, Sanborn & Love (2001)*, who observed high dropout rates and poor compliance in long-period exercise interventions. Additionally, in resistance exercise, longer intervention durations were also negatively associated with whole-body BMC, possibly reflecting a reduced bone sensitivity to prolonged mechanical loading. Prior research has suggested that adequate recovery intervals between exercise sessions enhance interstitial fluid flow and promote osteocyte network synchronization (*Gross et al., 2004*), both of which are critical for maximizing bone adaptation (*Robling, Burr & Turner, 2000*). Notably, no comparable time-dependent associations were observed at the lumbar spine or femoral neck, reinforcing the site-specific nature of bone adaptation to mechanical stimuli. These findings underscore the importance of optimizing not only exercise type and intensity but also intervention period and recovery strategies to maximize skeletal benefits in children and adolescents.

Notably, this meta-analysis innovatively compared the effects of high-impact jumping and resistance exercise on BMC in children and adolescents. It was found that high-impact jumping significantly increased BMC in key weight-bearing sites such as the lumbar spine and femoral neck. However, there are several limitations. First, the number of included studies is relatively small, and the studies on the two types of exercise are imbalanced. There are eight studies on jumping, but only four studies on resistance exercise, which may affect the accuracy of the results. Additionally, the included studies comprise four non-randomized controlled trials with uneven distribution: one non-randomized controlled trial among the eight jumping studies, and one randomized controlled trial among the four resistance exercise studies. Therefore, the imbalance in both the number of studies and the types of trials included may influence the results. Future research should include more high-quality studies and explore the potential synergistic effects of combined interventions.

## CONCLUSIONS

This study demonstrates that high-impact jumping significantly improves lumbar spine and femoral neck BMC in children and adolescents, with particularly pronounced effects observed in girls. In contrast, resistance exercise did not yield statistically significant improvements in BMC, possibly due to the limited number of studies and methodological

limitations. Future research should focus on high-quality randomized controlled trials to inform and optimize bone health interventions for children and adolescents.

### Funding
This study was supported by the Social Science Planning Research Program of Shandong Province in 2021 (21 DTYJ 03). The funders had no role in study design, data collection and analysis, decision to publish, or preparation of the manuscript.

### Grant Disclosures
The following grant information was disclosed by the authors:
The Social Science Planning Research Program of Shandong Province in 2021: 21 DTYJ 03.

### Competing Interests
The authors declare there are no competing interests.

### Author Contributions

- Tingting Miao analyzed the data, prepared figures and/or tables, authored or reviewed drafts of the article, and approved the final draft.
- Xun Li conceived and designed the experiments, authored or reviewed drafts of the article, and approved the final draft.
- Wenhua Zhang performed the experiments, prepared figures and/or tables, and approved the final draft.
- Fengying Yang performed the experiments, prepared figures and/or tables, and approved the final draft.
- Xiaoqiang Wang conceived and designed the experiments, authored or reviewed drafts of the article, and approved the final draft.

### Data Availability
    This is a systematic review/meta-analysis.

### Supplemental Information
Supplemental information for this article can be found online at http://dx.doi.org/10.7717/peerj.19616#supplemental-information.

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
