# Peer review of "Effects of high-impact jumping versus resistance exercise on bone mineral content in children and adolescents: a systematic review and meta-analysis"

_PeerJ, doi:10.7717/peerj.19616_

## Round 0.1 · original submission · Major Revisions

· Academic Editor

Major Revisions

Two reviewers gave substantial comments/suggestions on your manuscript. It looks like your rationale and methods have lots of flaws. If not carefully revised, you might overestimate your findings. Please read them carefully and revise it accordingly. I will make my decision based on your revision and comments/suggestions from two reviewers.

Reviewer 1 ·

Basic reporting

Insufficient background-
See "additional comments" for details.

Experimental design

Literature summary is insufficient, gap in the literature not identified and shown. Several methodological weaknesses and incomplete description of methodology. See "additional comments" for details.

Validity of the findings

Novelty and impact not discussed as the summary of existing literature and knowledge gaps is insufficient.

No sensitivity analyses on robustness.

Conclusions are overstating the findings. The small number of included studies and their quality needs to be discussed and included in conclusion.


See "additional comments" for details.

Additional comments

General comments
This meta-analysis examine the effect of impact, resistance and combined interventions on BMD and BMC in adolescents. Detailed comments are outlined below. My main concerns are around methodological rigor (pooling of different bone sites, not following PRISMA guidelines, search strategy) and overstating the findings when there the results are based on a small number of trials: beneficial effects of RT on FN BMC (3 trials), FN BMD (3 trials), WB BMC (2 trials). Please also provide a clear definition of “impact” or “jumping” or “high-impact jumping” exercise.
Abstract
- Conclusion: “High-impact jumping, resistance, and their combined exercises all promote BMC in adolescents, with resistance exercise significantly increasing femoral neck BMC and BMD in adolescents» This is not correct. There were no benefits of jumping or combined training. You cannot pool the results from the 3 different sites. Site-specificity is one of the basic principles of bone response and pooled estimates are therefore not only meaningless but also misleading.
Introduction
- Line 48: what do those 40% mean? At what age? For men and women?
- L56: This is not correct. Increased PBM does not delay the onset of menopause!
- L58-59: this sentence is contradicting. You say that only 90% of PBM are acquired before age 20 years but at the same time you say that PBM is reached at the end of puberty? Then in the next sentence (L61) you talk about accumulation of 50-60%. This section is very unclear and needs revision.
- L67: “response”? instead of “reconstruction”
- L74: It is not sufficient to state that “there are few meta-analyses on the topic”. Please carefully summarize existing meta-analyses (because there are some!) and justify why your meta-analysis is needed and which research gap it can address.
- L79: Not sure evidence from a meta-analysis like this will allow “personalized interventions” in the future.
Methods
- I disagree that it adheres to PRISMA guidelines. Please carefully check the PRISMA checklist and make sure you add all the criteria before stating it adheres to PRISMA guidelines. Just to list some examples, the following items on the PRISMA checklist are missing in your manuscript:
o 6. Specify the date when each source was last searched or consulted
o 7. Present the full search strategies for all databases, registers and websites, including any filters and limits used
o 10b. Describe any assumptions made about any missing or unclear information.
o 13b. Describe any sensitivity analyses conducted to assess robustness of the synthesized results
o 16b. Cite studies that might appear to meet the inclusion criteria, but which were excluded, and explain why they were excluded
o 24b. Indicate where the review protocol can be accessed, or state that a protocol was not prepared.
- The review was registered on PROSPERO on 11th December 2024, and all searches were completed in December 2024 which means registration was not prospective and there is no option to check whether changes to the initial protocol were made. Please state that in your manuscript .
- Line 91: provide the exact date.
- L97: it looks like this is the strategy used on PubMed. Please also describe the filters/limits and other restrictions (e.g., language) that were applied or state that none were applied.
- L97: for the search terms. Why did you only search for “high-impact” and not just “impact”. Did you consider searching for just “exercise” (MeSH) and then screen the articles for intervention type? Why did you not use MeSH terms for Bone? I am not entirely sure that this strategy was adequate to identify all relevant articles.
- “Participants” may be a more appropriate and ethical term than “subject”. I suggest replacing it throughout the manuscript.
- 106: you use the terms jumping and high-impact training. Please provide a clear definition of what you considered as jumping or impact training.
- 108: please clarify what you mean by “or the original exercise routine without additional exercise interventions”
- 111: according to several guidelines, justify why non-randomized trials were included and discuss how the inclusion of those could impact results.
- 122: this indicates that you used post-intervention values for meta-analysis. Please specify in the methods.
- 136-138: What are these threshold based on? Please provide a reference. I suggest using Cochrane ones which can be found here: 9.5.2 Identifying and measuring heterogeneity
- 140-141: please specify subgroup or sensitivity analyses

Results
- 150: “Among them, 8 interventions did not match the target group”. Please clarify, I do not understand this sentence. Do you mean that 8 trials were excluded due to wrong study population
- 152: “10 could not extract data” seems like a lot (10 trials). What was the issue?
- 171: The information provided in this paragraph is already shown in Figure 3. Instead of repeating all the information, I suggest focusing on details nor provided in the Figure. E.g., what “other risk of bias” you found
- 208: according to the Cochrane guidelines, 46% is considered moderate heterogeneity, actually bordering “substantial” (>50%) consider random-effect models.
- 210: You cannot pool the results from the three different measuring sites! Bone response depends on the assessment site and bone composition. This pooled results is meaningless and misleading and needs to be removed. In fact, the actual effects at each measuring site are not significant!
- 264: again, you cannot pool these results!
- 303: Assessment of publication bias is only recommended if a minimum of 10 trials are included. For each of the 3 outcome measures (LS, FN, WB) you have less than 10 trials. Please discuss that reliability is of publication assessment is limited.

Discussion
- The first paragraph is more like a literature review and belongs to the introduction. I recommend starting the discussion with a summary of the findings of the present meta-analysis.
- Line 323-325: This is the aim, phrased in future tense. It definitely does not belong to the discussion.
- You are comparing to several meta-analyses in different populations (e.g., Zhang 2024, 2025) and cite exercise guidelines for OP (Beck 2017). The primary focus of the discussion should be to compare to literature in adolescence.
- The discussion is long and not very concise. Instead of stating many findings from other studies, I suggest discussing your results. How do they align with the literature or why are they in contrast.
- 377: This is an important point. It would be nice to have a sensitivity analysis, excluding trials shorter than 6 months.
- Make sure you consider the methodological quality and the small number of trials included in your MA (e.g., 2 for RT) when discussing the results.
Conclusion
- Maybe also state that there were no effects of RT at LS and WB and there were no effects of jumping exercise.
- I recommend stating that results are limited by small number of included trials.
- The part 405-410 may be true but is nothing that was specifically assessed in your aims or analyses and thus I think it does not belong to your conclusions.
- I suggest aligning the main conclusion with the one in the abstract.

·

Basic reporting

The authors have conducted a systematic review and meta-analysis on the effect of high impact/jumping exercises and resistance training or their combination on adolescents’ bones measured with dual-energy X-ray absorptiometry (DXA). The authors concluded that the assessed exercises and their combination are effective, but they do not differ from each other in terms of the magnitude of treatment effect. Particularly, the benefits of jumping and high impact exercises for bones are well established in scientific literature and have been known since 90s and early 00s. Therefore, it remains a bone of contention whether further validation on this topic is truly needed.
Basically, the paper is well written, and the message is clear.

Experimental design

Research questions are clearly defined and relevant to what is known and what may not be known about the topic at present. The meta-analysis was performed lege artis on most aspects. For particular concerns, please see my specific comments concerning the validity of findings.

Validity of the findings

The effectiveness of high impact training and maximal resistance training, inducing high strain rates and magnitudes on loaded bones is well-established, and to this reviewer, no further validation is needed as claimed by the authors in Line 43. Be it noted that this claim is not reiterated in the main text. As the authors are likely aware, athletes provide natural long-term experiments to evaluate the effects of intensive jump/high impact training and heavy resistance training on bones at different anatomic sites. Athletes typically train systematically at high intensity as appropriate for the given sports and have started their training at early adolescence when the osteogenic potential of physical loading is maximized. The indirect evidence from athlete studies is thus so compelling that the authors should consider adding a paragraph on this topic to Discussion and cite relevant athlete studies. Obviously, the strength of evidence from athlete studies is compromised by the cross-sectional design (some prospective studies exist) and is thus subject to selection bias. On the other hand, the consistent findings from athletic unilateral training (e.g. using search words like racquet players side difference bone in PubMed) speak for the effectiveness of high impact loading on loaded bones.
I found the relevant literature used in the analysis incomplete, which therefore needs updating and reanalysis. Please see my specific comment concerning Line 91 below.
If only post-intervention mean data were considered as stated in Line 122, this approach is subject to fallacy and may mask or exaggerate the true training effects if the baseline mean values differed between groups. Unless the baseline levels in both groups are equal, the results based on post-intervention mean data can be fallacious. Typically, meta-analysis is based on comparisons of mean changes observed in groups. Please verify the appropriateness of your analysis and revise as appropriate. If mean changes are not available, the post-intervention mean values can be normalized by respective baseline data, as done by in a meta-analysis by Nicander R et al (BMC Med 2010).
The authors should also check if there is a difference in treatment effect between sexes, provided that relevant studies in male and female adolescents are available.
Since both randomized controlled trials (RCT) and non-randomized controlled trials (non-RCT) were analyzed, the authors may present their findings also separately. It is known from scientific literature that the latter tend to show somewhat greater treatment effects.
The authors should consider whether aBMD adds to knowledge compared to what BMC data tells. Please see my specific comment on Line 322.

Additional comments

Since all (apparently this was the case?) studies included were performed using DXA, proper acronyms should be used: BMC for bone mineral content and aBMD for areal bone mineral density. For clarity, the expression “DXA measured” may sometimes be written before the acronyms.
Whenever bone is measured with DXA, it is advisable to use the acronym aBMD (areal bone mineral density) and always mention the skeletal site. Please check the whole manuscript.

Specific comments
Line 32. Please mention the skeletal site.
Line 75. Please provide references for meta-analytic studies mentioned in the text.
Line 91. Please specify the start years of the literature searches. The reason why I'm asking is that some relevant studies may be missing, including the controlled trial of premenarcheal and postmenarcheal girls by Heinonen A et al. (Osteoporosis International 2000). Please confirm that all relevant literature is considered in the analysis
Line 104. What was the rationale for choosing 10 years as the lower limit for being included in the analysis? Does this comply with the definition of an adolescent? Please clarify.
Line 106. Organization et al. Is this reference properly written? Please check.
Line 133. Please clarify how the clinical relevance of the effect size was interpreted.
Lines 260 and 265. This is confusing. What skeletal site? aBMD values measured from different sites, cannot be pooled (averaged) to a single analysis. aBMD from different sites are not additive. The analyses should be performed only site-specifically. The first "general" paragraph should be revised.
Line 311. Under loading circumstances generated by muscle activity (e.g. during jumping) bones are basically subject to compressive and shear forces. This is because the concurrent muscle activity tends to direct the loading to bones in a way that tension is avoided (bone is weaker in tension than in compression). I therefore suggest rephrasing these sentences. A safe way is to talk about physical or mechanical loading instead of specific mechanical concepts.
Line 322. aBMD is what is basically measured with DXA, while BMC represents the integral over the area considered bone according to certain thresholds. Thus, BMD and BMC are collinear and provide theoretically the same information if the measured bone projection remains unchanged. The same bone scanned from a different angle can yield a fallacious aBMD value. True measurable increases in bone width may occur during axial growth if the follow-up is sufficiently long. The authors may consider these lines of thought when presenting their data. BMC is a tangible measure and reflects changes in bone mass.
Lines from 328 to 394. Please make it clear to the reader which study and in what group and skeletal site is in question. Also, please discuss only RCT studies that have shown statistically significant training effects. Otherwise, the purpose of these paragraphs remains obscure, at least to this reviewer. Revising and sharpening this part is strongly recommended.
Line 375. I wonder whether the effect of the intervention duration shows any association with changes in bone. The authors may consider conducting a meta-regression analysis. Later in Lines 399 and 400, the authors state that the influence of duration, frequency, and intensity of the intervention was not evaluated. Please clarify why this was not done.

---

## Round 0.2 · Minor Revisions

· Academic Editor

Minor Revisions

After carefully reviewing your revised version, one of the PeerJ Section Dditors suggested that Robumeta approach can make your meta-analysis better. Please refer to following page: https://arxiv.org/abs/1503.02220

Please revise it accordingly. After receiving your revision, I will make my last decision.

---

## Round 0.3 · accepted · Accept

· Academic Editor

Accept

Thank you for the revision. I will accept this manuscript and recommend it for publication in PeerJ.